# Modified Biochanin A Release from Dual pH- and Thermo-Responsive Copolymer Hydrogels

**DOI:** 10.3390/polym13030426

**Published:** 2021-01-29

**Authors:** Ivana Gajić, Snežana Ilić-Stojanović, Ana Dinić, Aleksandar Zdravković, Ljiljana Stanojević, Vesna Nikolić, Ljubiša Nikolić

**Affiliations:** 1Faculty of Technology, University of Niš, Bulevar Oslobodjenja 124, 16000 Leskovac, Serbia; ivana@tf.ni.ac.rs (I.G.); anatacic@tf.ni.ac.rs (A.D.); stanojevic@tf.ni.ac.rs (L.S.); nikolicvesna@tf.ni.ac.rs (V.N.); nljubisa@tf.ni.ac.rs (L.N.); 2Department of Technology and Art Studies, Academy of Professional Studies South Serbia, Leskovac, Vilema Pušmana 17, 16000 Leskovac, Serbia; zdravkovic.aleksandar87@gmail.com

**Keywords:** poly(*N*-isopropylacrylamide-co-acrylic acid), hydrogel, swelling kinetics, lyophilization, residual reactants, biochanin A

## Abstract

The temperature- and pH-responsive poly(*N*-isopropylacrylamide-co-acrylic acid), p(NIPAM-co-AA), copolymer was synthesized by free radical polymerization and examined as a carrier for modified release of biochanin A. Biochanin A is a biologically active methoxylated isoflavone which exhibits estrogenic and other pharmacological activities. Due to its poor aqueous solubility and extensive first-pass metabolism, biochanin A has low bioavailability. The aim of this work was to incorporate biochanin A into the synthesized p(NIPAM-co-AA) copolymer and to examine its release at the body temperature and pH values that correspond to pH values of vaginal and rectal cavities. The amount of released biochanin A was monitored by the ultra-visible spectroscopy (UV-Vis) method. The structure of synthesized p(NIPAM-co-AA) copolymer and copolymer with incorporated biochanin A were characterized by using Fourier transform infrared spectroscopy (FTIR) and scanning electron microscopy (SEM) methods. The content of residual monomers in the synthesized copolymer was analyzed by using the high-pressure liquid chromatography (HPLC) method. The swelling behavior of p(NIPAM-co-AA) copolymer was monitored in relation to the temperature and pH values of the surrounding medium. For modelling the process of p(NIPAM-co-AA) copolymer swelling, the full three-level factorial design was applied.

## 1. Introduction

Biochanin A is a methoxylated isoflavone which can be found predominantly in plants of the Fabaceae family, primarily in red clover (*Trifolium pratense* L.), soybean (*Glycine max* L.), alfalfa (*Medicago sativa* L.) and chickpea (*Cicer arietinum* L.) [1]. The chemical structure of biochanin A is given in Figure 1. Due to its structural similarity to human estrogen, biochanin A can exhibit both estrogenic and antiestrogenic activity depending on the applied concentration [2]. Also, Biochanin A has other pharmacological activities, such as anti-inflammatory, antioxidant, antimicrobial, antidiabetic, osteoprotective and anticancer activity, primarily against hormone-dependent cancers [3,4,5,6,7,8].

The oral administration of biochanin A is primarily limited due to its poor aqueous solubility [9]. Further, pharmacokinetic studies have shown that biochanin A undergoes extensive first- and second-pass metabolism despite efficient penetration through the enterocyte membrane. Due to conjugation and hydrolysis by enzymes and intestinal bacteria, biochanin A undergoes extended enterohepatic recirculation. In this way, the biochanin A ability to enter the systemic circulation is further decreased as well as its bioavailability [10]. Since biochanin A can inhibit P-glycoprotein (P-gp), as well as other transport proteins in the liver and intestines, and represents a potential substrate in glucuronidation and sulfation reactions, biochanin A administered per os can enter into unpredictable interactions with various drugs [11]. The absorption rate of biochanin A is limited by its dissolution rate, so biochanin A is classified as a BSC (The Biopharmaceutics Classification System) Class II drug [12].

In order to improve the physico-chemical characteristics and safety of biochanin A, as well as to avoid its interactions with other drugs, it is necessary to develop novel drug delivery systems for biochanin A. Wu et al. [13] have prepared polymeric mixed micelles with Pluronic 127 and Plasdone S630 in a ratio of 1:1 as carriers for biochanin A. By incorporating biochanin A into mixed micelles, a slower release was achieved during 72 h, with total released amount of 72.86% ± 7.28%, compared to free biochanin A which was released rapidly (66.84% ± 6.68%) within the initial 8 h. These results indicated that mixed micelles as carriers for biochanin A increase the solubility and absorption of orally administered biochanin A and consequently its bioavailability by 2.16 times [13].

Sachdeva et al. [14] have described the preparation of enteric-coated microparticles with biochanin A as a novel drug delivery system. About 84% of the biochanin A was released from the microparticles within 48 h at pH 6.8, and only 20% of the biochanin A at pH 1.2. The pharmacokinetic studies have shown that incorporation of biochanin A into microparticles prolongs blood circulation, reduces clearance and enhances oral bioavailability of biochanin A [14].

Tao et al. [15] have used solid lipid nanoparticles as carriers for biochanin A. The release of biochanin A from solid lipid nanoparticles was monitored in vitro by using the dialysis bag method in phosphate buffer solution pH 7.4 at 37 ± 0.5 °C. The released amount of free biochanin A was 97% within the initial 8 h, while the release of incorporated biochanin A from solid lipid nanoparticles was prolonged, with 95% of the biochanin A released after 48 h [15].

Nanostructured lipid carriers with a biphasic release pattern were also used as a delivery system for biochanin A. The active substance on the surface of nanoparticles and in the outer shell was released abruptly during the initial 4 h. The release rate from the nanoparticles’ cores was relatively constant. Incorporation of biochanin A into nanostructured lipid carriers resulted in increased absorption after oral administration, reduced first-pass metabolism and seven times higher bioavailability of the biochanin A [16].

In order to avoid extensive first-pass metabolism, the carriers for alternative routes of biochanin A administration were also developed [17]. Hanski et al. have prepared buccal film formulation with biochanin A using water-soluble polymers (hydroxypropyl cellulose and hydroxypropyl methylcellulose), plasticizers and disintegrants. More than 90% of biochanin A was released from the film formulation during 4 h. The advantages of this formulation type with water-soluble polymers can include the reduction of applied dose of active substance by increasing its bioavailability and entrance into the systemic circulation, without undergoing first-pass metabolism in liver [17]. 

The similarity of hydrogels to living tissue creates many possibilities for their application in biomedicine [18,19,20]. Many patents and scientific papers on the possible applications of hydrogels in drug delivery have been published, but only a few of them have resulted in commercial products. Hydrogels are interesting as drug delivery systems due to their unique physical properties [21]. The high porosity that characterizes hydrogels can be easily adjusted by controlling the crosslinking density within the matrices, and thus their affinity for water. Due to the porous structure of hydrogels, drugs can be incorporated into the matrices and then released under certain conditions. The advantages of hydrogels as drug carriers include the possibility of prolonged, continuous release, which results in maintaining a high local concentration of the pharmaceutically active substance over a long period of time [22].

The active substance can be incorporated into the hydrogel by hydrogel swelling to the equilibrium in the active substance solution. The hydrogel can be designed to swell and release active substance in the specific environment depending on the conditions, such as pH, ionic strength, temperature, etc. An increase of body temperature leads to the hydrogel contraction, and subsequently to the active substance release [23]. The incorporated drug can be released from hydrogel through several mechanisms: diffusion controlled, swelling controlled, controlled chemically and environmentally [18].

In this paper, poly(*N*-isopropylacrylamide-co-acrylic acid) polymeric carrier, p(NIPAM-co-AA), for alternative application routes of biochanin A, was synthesized. The copolymer p(NIPAM-co-AA) has specific properties due to the presence of carboxyl and amino groups in its structure, which makes it suitable for controlled drug release [24]. This type of polymer has the ability to absorb large amounts of water or physiological fluids (up to 2000 times larger than its mass) [25]. It also reacts to external stimuli, primarily to pH and temperature, which changes its absorption properties. The behavior of the copolymer at the certain pH value depends on the pKa and pKb values of the acidic and basic functional groups in the copolymer. At pH values lower than pKa, the acrylic acid carboxyl groups are protonated (COOH) and the hydrogel is contracted. At lower pH values, the amino groups in the NIPAM monomer are protonated (NH_2_^+^) and electrostatic repulsive forces increase the hydrophilicity of the polymeric network and hydrogel swells [26]. Hydrophilic acrylic acid contributes to a higher degree of water absorption and to a higher value of lower critical solution temperature, which is close to the physiological body temperature [27]. The listed characteristics of the p(NIPAM-co-AA) copolymer indicate that it can be used as a suitable carrier for drug delivery [28].

## 2. Materials and Methods 

### 2.1. Materials

*N*-isopropylacrylamide (NIPAM; purity 99%) and 2,2’-azobis(2-methylpropionitrile) (AIBN; purity 98%) were purchased from Acros Organics (Morris Plains, NJ, USA), while acrylic acid (AA; purity 98%) and ethylene glycol dimethacrylate (EGDM; purity 97%) were purchased from Fluka Chemical Corp (Buchs, Switzerland). Biochanin A was bought from Sigma Aldrich (Steinheim, Germany). Methanol, 99.9% (Sigma-Aldrich GmbH, Steinheim, Germany), ethanol, 96% (Zorka Pharma, Šabac, Serbia), potassium bromide (KBr; spectroscopic purity) (Merck, Darmstadt, Germany), hydrochloric acid (HCl; ≥36.5%), acetone, 99.5%, and sodium hydroxide (NaOH) (Centrohem, Belgrade, Serbia) were also used.

### 2.2. Synthesis of the Copolymeric p(NIPAM-co-AA) Hydrogel

The copolymeric hydrogel of poly(*N*-isopropylacrylamide-co-acrylic acid), p(NIPAM-co-AA), was synthesized by radical polymerization of NIPAM and AA (5 mol%) monomers using 1.5 mol% of EGDM as a cross-linker. The polymerization reaction was initiated by adding 2.7 mol% of AIBN. Acetone was used as a solvent. After dissolving the reactants, the homogenized reaction mixture was injected into the glass tube, being sealed afterwards. The polymerization reaction was performed in the following temperature regime: 0.5 h at 75 °C, 2 h at 80 °C and 0.5 h at 85 °C. After cooling, the long cylinder of synthesized copolymer was cut into disks (*d* × *h* = 5 × 2 mm, where *d* is the disk diameter and *l* is the thickness after drying in mm). The synthesized copolymer was treated for 72 h with methanol (60 cm^3^ of methanol per 1 g of copolymer) to remove residual reactants. The treated copolymer was rinsed using solutions of methanol/distilled water in a ratio of 75%/25%, 50%/50%, 25%/75% and 0%/100% in order to remove methanol and then dried at 40 °C to constant weight. The decanted methanol solution was analyzed in order to determine residual reactants content by using high-pressure liquid chromatography (HPLC). The obtained copolymer was used for the incorporation of biochanin A.

### 2.3. Lyophilization of the Copolymeric p(NIPAM-co-AA) Hydrogel

Lyophilization of the p(NIPAM-co-AA) hydrogel in swollen state was performed on the device LH Leybold Heraeus, Lyovac GT2 (Frekendorf, Switzerland). The hydrogel was first frozen at –40 °C for 24 h. In the primary drying phase, the amount of the solution was reduced by sublimation at –30 °C and at the pressure of 5 Pa during 12 h. In the secondary drying phase (isothermal desorption), the hydrogel was heated at 20 °C and at the pressure of 5 Pa. The lyophilized hydrogel was stored at 4–8 °C and used for the incorporation of biochanin A.

### 2.4. Incorporation of the Biochanin A into the Copolymeric p(NIPAM-co-AA) Hydrogel

The solution of biochanin A (2 mg/cm^3^) was prepared by dissolving biochanin A in ethanol (96%, *v*/*v*). The mass of 0.020 g of the synthesized lyophilized and non-lyophilized p(NIPAM-co-AA) xerogel was weighed. The samples were poured with 0.6 cm^3^ of the biochanin A solution and allowed to swell for 4 h. The available amount of biochanin A for incorporation into the copolymer was 60 mg/g_xerogel_. After reaching equilibrium, the swollen p(NIPAM-co-AA) hydrogels with incorporated biochanin A were separated from the solution by decanting. The hydrogel samples were washed using distilled water to remove excess biochanin A. The mass of samples with incorporated biochanin A was measured in order to calculate the loading efficiency. 

The content of the incorporated biochanin A in the synthesized copolymers (lyophilized and non-lyophilized) was determined by measuring the mass of samples before and after swelling in the biochanin A solution. The loading efficiency (*η*) of biochanin A was calculated using Equation (1):(1)$$\eta \left(\%\right)=\frac{L_{g}}{L_{u}}\cdot 100$$
wherein *L_g_* is the mas of biochanin A incorporated into the hydrogel (mg/g_xerogel_) and *L_u_* is the initial mass of biochanin A in the swelling solution (mg/g_xerogel_).

### 2.5. Modified Release of Biochanin A from the Copolymeric p(NIPAM-co-AA) Hydrogel

The swollen lyophilized and non-lyophilized p(NIPAM-co-AA) hydrogels with incorporated biochanin A were poured with 2 cm^3^ of the adequate medium: a solution of hydrochloric acid pH 4.5 or a solution of sodium hydroxide pH 7.9. The samples were stirred and thermostated in a water bath at 37 °C for 4 h. The release of biochanin A was monitored by sampling 100 μL of solution over time (0, 0.5, 1, 2 and 4 h) and diluting with ethanol (96%, *v*/*v*) to the volume of 2 cm^3^. The absorbance of the prepared samples was determined at 261 nm by using the ultra-visible spectroscopy (UV-Vis) method.

### 2.6. Characterization Methods

#### 2.6.1. Fourier Transform Infrared Spectroscopy (FTIR)

The synthesized p(NIPAM-co-AA) xerogel and xerogel with incorporated biochanin A were ground to powder in an amalgamator (WIG-L- Bug, Dentsply RINN, a Division of Dentsply International Inc., York, PA, USA). FTIR spectra of the biochanin A, monomer NIPAM and xerogels were recorded by a technique of thin transparent pastilles by vacuuming and pressing under the pressure of about 200 MPa. The pastilles were prepared by mixing 150 mg of KBr and 1 mg of the sample. The comonomer AA was recorded as a thin film between two plates of zinc selenide (ZnSe). FTIR spectra were recorded in the area of wavenumbers from 4000 to 400 cm^–1^ on a Bomem Hartmann and Braun MB-series FTIR spectrophotometer (Hartmann & Braun, Baptiste, Quebec, QC, Canada). The obtained spectra were analyzed using the Win-Bomem Easy software.

#### 2.6.2. High-Pressure Liquid Chromatography (HPLC)

The HPLC method was applied for qualitative and quantitative analysis of the residual reactants content in the synthesized p(NIPAM-co-AA) copolymer. The combined methanol extracts obtained after processing the synthesized copolymer were used for the analysis. The analysis was performed on an Agilent 1100 Series HPLC device (Waldborn, Germany) with a Zorbax Eclipse XDB-C18 column, 250 × 4.6 mm, 5 μm (Agilent Technologies, Inc., Santa Clara, CA, USA) at 25 °C. Methanol was used as the mobile phase with a flow rate of 1 cm^3^/min. The injected sample volume was 10 μL. The detection was performed on a Diode array detector (DAD) 1200 Series detector at wavelengths of 205 nm for AA and EGDM, and 220 nm for NIPAM. For the construction of calibration curves, the series of adequate standard solutions of known concentrations were prepared. All samples were filtered on the Econofilter with the pore diameter of 0.45 μm and used for the HPLC analysis. The recorded spectra were processed using Agilent ChemStation software. Based on the constructed calibration curves, the equations for determining the content of NIPAM, AA and EGDM in the combined methanolic extracts obtained by processing the synthesized p(NIPAM-co-AA) copolymer were obtained. The calibration curve for NIPAM was linear in the concentration range of 0.005–0.506 mg/cm^3^. Equation (2) with the correlation coefficient *R*^2^ = 0.997 applies:(2)A=25,985.512·c+266.829

The calibration curve for AA was linear in the concentration range of 0.010–0.300 mg/cm^3^. Equation (3) with the correlation coefficient *R*^2^ = 0.989 applies:(3)A=31,560.568·c−75.877

The calibration curve for EGDM was linear in the concentration range of 0.005–0.264 mg/cm^3^. Equation (4) with the correlation coefficient *R*^2^ = 0.989 applies:(4)A=48,598.866·c+673.254

In the Equations (2)–(4), *A* is the peak area (mAU⋅s) and *c* is the content of NIPAM, AA and EGDM (mg/cm^3^), respectively. From the peaks’ integration data of the tested methanol extract samples, the obtained peak area values were in the range of the calibration curve.

#### 2.6.3. Swelling Study

The lyophilized and non-lyophilized p(NIPAM-co-AA) xerogels were immersed in the solutions of certain pH values (3.5, 6.0 and 8.5) and the swelling process was monitored gravimetrically. The solutions of the given pH values were prepared using HCl or NaOH and the acidity was measured using a digital pH meter (HI9318-HI9219, HANNA, Woonsocket, RI, USA). The hydrogel samples were taken out from the solutions and the excess solution was removed from their surface. The sample mass was measured in certain periods of time until the equilibrium was reached, i.e., until the constant mass of hydrogel. The swelling degree, *α*, was calculated according to Equation (5):(5)$$\alpha =\frac{m-m_{0}}{m_{0}}$$
where *m*_0_ is the mass of the dry gel and *m* is the mass of swollen hydrogel in the moment of time, *t*.

Equation (6) is applied to analyze the nature of the diffusion process of the solvent within the hydrogel matrix. This equation is valid for the initial phase of the swelling (*M_t_*/*M_e_* ≤ 0.6) [29,30]:(6)$$F=\frac{M_{t}}{M_{e}}=kt^{n}$$
where *F* is a fractional sorption, *M_t_* is the mass of the absorbed solvent at the time *t*, *M_e_* is the mass of the absorbed solvent in the equilibrium state, *k* is the constant characteristic for a certain type of polymer network (min^1/n^) and *n* is the diffusion exponent. By taking the logarithm of Equation (6), Equation (7) is obtained:(7)$$\ln F=\ln\left(\frac{M_{t}}{M_{e}}\right)=\ln k+n\ln t$$

The values of the diffusion exponent *n* and constant *k* can be determined from the slope and intercept respectively, of the linear relationship between ln*F* and ln*t*.

The mechanism of the solvent diffusion is determined by the value of the diffusion exponent *n*. For the hydrogels with planar geometry at the value *n* = 0.5, the fluid transport mechanism corresponds to the Fickian diffusion mechanism (Case I), and the polymer chain relaxation rate is much higher than the diffusion rate. “Less Fickian” diffusion is the mechanism of the solvent diffusion at *n* < 0.5, and the solvent transport in the polymeric matrix is considerably slower than the relaxation of polymer chains. The hydrogel swelling can be controlled by both solvent diffusion into the matrix and the relaxation of polymer chains (0.5 < *n* < 1), which corresponds to non-Fickian diffusion (anomalous diffusion mechanism). If *n* = 1, the solvent diffusion process is much faster than the relaxation of polymer chains (Type II, Case II). If *n* > 1, the hydrogel swelling is also controlled by the polymer chains’ relaxation (Type III, Case III, Super Case II) [29,30,31,32,33,34].

Besides the mechanism of the solvent absorption, it is necessary to determine the solvent molecule diffusion coefficient (*D*). The most commonly used method for determining the diffusion coefficient, *D*, taking into account only the initial phase of swelling during which the thickness of the sample basically remains constant [34], is presented as Equation (8):(8)$$\frac{M_{t}}{M_{e}}=4{\left(\frac{Dt}{\pi l^{2}}\right)}^{1/2}$$
where *D* is the diffusion coefficient (cm^2^/min) and *l* is the thickness of the dry hydrogel (cm). By taking the logarithm of Equation (8), Equation (9) is obtained (the linear relationship between ln(*M_t_/M_e_*) and ln*t*):(9)$$\ln\left(\frac{M_{t}}{M_{e}}\right)=\ln\left(\frac{4D^{1/2}}{\pi^{1/2}l}\right)+\frac{1}{2}\ln t$$

The diffusion coefficient, *D,* can be calculated from the intercept of the linear relationship between ln(*M_t_*/*M_e_*) and ln*t*.

#### 2.6.4. Modelling the Process of p(NIPAM-co-AA) Copolymer Swelling 

The experimental design is a structured and organized way of conducting and analyzing controlled experiments in order to evaluate the factors (independent variables, *X*) that affect the system response (dependent variable, *Y*) [35]. For modelling the process of p(NIPAM-co-AA) hydrogels swelling, the full three-level factorial design was applied. In full factorial design, the effects of all experimental factors and their interaction effects on the system response are investigated [36]. The system response is the equilibrium swelling degree (*α*_e_) of the p(NIPAM-co-AA) copolymeric hydrogels. Factors and levels applied in the three-level factorial design are given in Table 1.

The analysis of variance (ANOVA) test was used for selection and evaluation of the model adequacy and statistically significant factors in the model. The factors and interactions with values of probability levels (*p*) lower than 0.05 were considered as statistically significant members. The optimization of the swelling process of p(NIPAM-co-AA) hydrogels was performed by using Design-Expert^®^ software, version 7.0.0 (Stat-Ease Inc., Minneapolis, MN, USA).

#### 2.6.5. UV/Vis Spectrophotometry

The absorbances of the biochanin A samples for construction of the calibration curve and monitoring the modified release from p(NIPAM-co-AA) hydrogels (lyophilized and non-lyophilized) were measured at 261 nm in the quartz cuvette (1 × 1 × 4.5 cm) on a Varian Cary-100 spectrophotometer (Mulgrave, Victoria, Australia) at room temperature. UV spectra were processed using the Cary WinUV software. Ethanol (96%, *v*/*v*) was used as a blank. 

The calibration curve was constructed as the dependence of the absorbance at 261 nm on the known concentration of the biochanin A. The stock solution of the biochanin A (2 mg/cm^3^) was prepared by dissolving biochanin A in ethanol (96%, *v*/*v*) and then diluted with ethanol in the concentration range of 2–10 μg/cm^3^. The obtained calibration curve for biochanin A with the correlation coefficient (*R*^2^) of 0.999 is given in Equation (10):(10)A=0.1884·c+0.0202

#### 2.6.6. Scanning Electron Microscopy (SEM)

Scanning electron microscopy (SEM) was used to examine the morphology of the synthesized copolymeric p(NIPAM-co-AA) hydrogel. The lyophilized samples of the synthesized p(NIPAM-co-AA) copolymer and copolymer with incorporated biochanin A in the equilibrium swelling state were lyophilized on an Edwards Mini Fast 680 laboratory freeze-dryer (Edwards Ltd, Eastbourne, UK). The lyophilized samples were immersed into nitrogen before cutting to prevent breakage and deformation. After that, the samples were sprayed by an alloy of gold and palladium (85%/15%) under vacuum in a Fine Coat JEOL JFC-1100 Ion Sputter (JEOL Ltd., Tokyo, Japan). The metalized samples of p(NIPAM-co-AA) were scanned with a JEOL Scanning Electron Microscope JSM-5300 (JEOL Ltd., Tokyo, Japan).

## 3. Results and Discussion

### 3.1. Synthesis of the Poly(N-Isopropylacrylamide-co-Acrylic Acid) Polymer

Simultaneous reaction to various external stimuli, such as temperature and pH of the surrounding medium, is one of the important conditions for hydrogels’ application, especially as drug carriers [37,38,39]. In order to obtain temperature- and pH-responsive hydrogel, the NIPAM monomer was copolymerized with the ionic monomer AA using EGDM as a cross-linker. The reaction of free radical polymerization was initiated by 2-cyano-2-propyl radical formed at high temperature by degradation of the initiator 2,2’-azobis(2-methylpropionitrile) (Figure 2).

The resulting primary radicals in the initiation phase react with monomer and crosslinking molecules to form radical species, whose structures are shown in Figure 3.

In the propagation and termination phases of the polymerization reaction, a cross-linked structure of the p(NIPAM-co-AA) copolymer was formed. The possible structure of the copolymeric hydrogel network is shown in Figure 4.

The synthesized p(NIPAM-co-AA) hydrogel was characterized using the FTIR method, the content of the residual reactants was determined and the swelling degree as a function of pH and temperature was examined.

### 3.2. Structural Characterization of Synthesized Poly(N-Isopropylacrylamide-co-Acrylic Acid) Copolymer

#### 3.2.1. FTIR Spectroscopy Analysis

The FTIR spectrum of the NIPAM monomer is given in Figure 5. The amide band I assigned to the valence vibrations of the C=O group is expected at 1680 cm^−1^. Due to conjugation and the pronounced negative inductive effect of the vinyl C=C bond on the carbonyl group, this band occurs at the maximum of 1658 cm^−1^ in the spectrum of NIPAM. The N–H valence vibrations of the secondary amides give one band in the range of 3500–3100 cm^−1^, which appears as a strong intensity band with maximum at 3297 cm^−1^. The amide band II is a result of the N–H in-plane deformation vibrations and falls in the area of the amide band I, so it is not visible in the NIPAM spectrum. The amide band III derived from the C–N valence vibrations coupled with the N–H deformation vibrations appears at 1246 cm^−1^. The isolated vinyl hydrogen atom (=C–H) gives only one band with maximum at 3072 cm^−1^, which is a result of asymmetric vibrations of the vinyl group, ν_as_(=C–H). The number and position of bands originating from out-of-plane bending vibrations of the vinyl C–H bond, γ(=C–H) depends on the degree of double bond substitution. Two bands originating from γ(=C–H) vibrations at 989 and 918 cm^−1^ indicate a monosubstituted double bond. Conjugation of the vinyl group with the carbonyl group of the secondary amide shifts the vibration frequency of the C=C bond, ν(C=C), to lower values, while polarization increases the intensity of C=C absorption, which results in a strong-intensity band at 1619 cm^−1^. 

The FTIR spectrum of the comonomer acrylic acid is shown in Figure 6.

In the FTIR spectrum of AA (Figure 6), a wide absorption band originating from O–H valence vibrations, ν(OH), of the carboxyl group is observed in the range of wavenumbers from 3500 to 3200 cm^−1^, which is in accordance with the literature [40]. In this area, the band originating from the isolated vinyl hydrogen atom (=C–H) is also expected but difficult to observe due to the overlapping with the band from O–H valence vibrations. This band gives a weak maximum at 3067 cm^−1^. The absorption band in the range of wavenumbers from 1710 to 1690 cm^−1^ in the spectra of aliphatic carboxylic acids is assigned to valence vibrations of C=O groups, ν(C=O). In the spectrum of AA, it appears as a strong-intensity band with a maximum at 1702 cm^−1^. Due to conjugation with the vinyl group, this band is shifted to lower wavenumbers. For the same reason, the band of valence vibrations of C=C bond from vinyl group, ν(C=C), is shifted to lower wavenumbers and it appears as a strong-intensity band with a maximum at 1614 cm^−1^. The valence vibrations of C–O bond, ν(C–O), coupled with in-plane deformation vibrations, δ(OH), give two bands in the spectrum of AA with maxima at 1433 and 1241 cm^−1^, and confirm the presence of the COOH group. The characteristic bands originating from out-of-plane bending vibrations of the vinyl C–H bond appear at 1044 and 982 cm^−1^. The presence of these bands in the spectrum of AA indicates that the double bond is monosubstituted. 

The FTIR spectrum of the synthesized copolymer p(NIPAM-co-AA) with 5 mol% of acrylic acid and 1.5 mol% of the cross-linker EGDM is shown in Figure 7.

In the FTIR spectrum of the synthesized p(NIPAM-co-AA) copolymer (Figure 7), the absence and shifts of certain characteristic absorption bands of NIPAM and AA can be observed. The absence of the absorption bands originating from the valence vibrations of vinyl C=C bonds, ν(C=C), of the monomers and cross-linker in the range of 1640–1620 cm^−1^, and in-plane deformation vibrations, δ(=C–H), in the range of 1450–1200 cm^−1^ indicates the successful polymerization. Also, the absence of the bands originating from out-of-plane C–H bending vibrations, γ(=C–H), which occur in the NIPAM and AA spectra at 989 and 918 cm^−1^, and 1044 and 982 cm^−1^ respectively, clearly indicates that vinyl groups of monomers participated in the polymerization process. The carboxyl and alkylated amide groups from the AA and NIPAM monomers respectively, are preserved in the structure of the copolymer, which is indicated by the presence of the corresponding bands in the FTIR spectrum of the copolymer. The broad absorption band in the FTIR spectrum of the copolymer has two saddles, one at 3488 cm^−1^ assigned to OH valence vibrations, ν(OH), from the carboxyl group of the AA comonomer, and the other at 3298 cm^−1^ assigned to N–H valence vibrations, ν(N–H), from the NIPAM monomer [41]. The absorption band with the maximum at 1720 cm^−1^ is assigned to valence C=O vibrations of the carboxyl group of the AA comonomer. The maximum of this band is shifted towards higher wavenumbers by 18 units relative to the same band in the FTIR spectrum of AA comonomers (Figure 6). In the FTIR spectrum of the copolymer, there is an absorption band at 1653 cm^−1^ which corresponds to the amide band I originating from the valence vibrations, ν(C=O), and is shifted by 5 units towards lower wavenumbers in relation to the same band in the FTIR spectrum of NIPAM (Figure 5). The analysis of the FTIR spectra of monomers and copolymer indicates that polymerization was achieved and that the assumed structure of copolymer p(NIPAM-co-AA) (Figure 4) is accurate.

#### 3.2.2. Residual Reactant Analysis

The HPLC method was used to analyze methanol solutions from the obtained p(NIPAM-co-AA) copolymer in order to determine the amount of residual unreacted monomers and the cross-linker. Under the selected chromatographic conditions, the retention time (*R*_t_) of 3.278 min corresponds to NIPAM, *R*_t_ = 3.082 min to AA, and *R*_t_ = 3.403 min to EGDM. The HPLC chromatograms and UV spectra of the monomers and cross-linker are shown in Figure 8a (I and II), b (I and II) and c (I and II), respectively. 

The unreacted amounts of monomers and cross-linker from the p(NIPAM-co-AA) copolymer synthesis calculated in relation to the total mass of the synthesized xerogel, as well as in relation to their initial amount in the reaction mixture, are presented in Table 2.

The obtained residual reactants content was within the acceptable limits and indicates almost complete conversion of the initial compounds in the process of p(NIPAM-co-AA) copolymer synthesis. The total content of the residual reactants is less than 1%, which is in accordance with the permitted limits for the similar materials [42]. Since the toxicity of residual reactants depends on their content, the synthesized copolymer p(NIPAM-co-AA) can be considered safe for use as a carrier of active substances.

#### 3.2.3. Swelling Study

In order to achieve the safe application of p(NIPAM-co-AA) hydrogels, the stability of NIPAM microgels with different content of AA under the various conditions of temperature, pH and sodium chloride concentrations were examined [43]. It has been shown that the increase of the temperature and sodium chloride concentration, as well as the decrease of pH, cause aggregation and reduce the stability of the microgel. The p(NIPAM-co-AA) microgel is unstable at high sodium chloride concentration, at temperature higher than 45 °C and at pH lower than 2.25 [43]. Taking into account the stability of p(NIPAM-co-AA) microgels and previous research on p(NIPAM-co-AA) hydrogels [44,45], the swelling study of the synthesized p(NIPAM-co-AA) hydrogel as a potential carrier for alternative routes of administration was conducted in the temperature range of 25–37 °C and at pH of 3.5–8.5, which correspond to temperature and pH of vaginal and rectal cavities.

The changes in the swelling degree of poly(NIPAM-co-AA) hydrogel before and after lyophilization, in function of time, in the solvents with different pH values (3.5 and 8.5) and temperatures (25 and 37 °C), are shown in Figure 9 and Figure 10.

The sample of the p(NIPAM-co-AA) hydrogel was swollen to equilibrium in distilled water and lyophilized to obtain a polymer with large pore size. Therefore, the lyophilized p(NIPAM-co-AA) hydrogel absorbed solvent rapidly and had higher swelling degree in the initial phase of swelling (Figure 9). It can be observed (Figure 9) that lyophilized p(NIPAM-co-AA) hydrogel reached lower values of the equilibrium swelling degree at both pH values (3.5 and 8.5), which could be the consequence of reduced flexibility of the polymer chains and lower solvent absorption during re-swelling. Both non-lyophilized and lyophilized p(NIPAM-co-AA) hydrogels had the significantly greater values of the equilibrium swelling degree in the alkaline medium at pH 8.5 (269.323 and 259.218) than in the acidic medium at pH 3.5 (11.531 and 11.226). The increase of pH value caused the expansion of the polymer network due to electrostatic repulsion between numerous ionized carboxyl groups (COO^−^) of the polymer chains [46]. 

By comparative analysis of the swelling process of both non-lyophilized and lyophilized p(NIPAM-co-AA) hydrogels at 37 °C (Figure 10), asimilar behavior of polymers was noticed at 25 °C. The polymer network that had already reached equilibrium was rigid after the lyophilization, and the lyophilized p(NIPAM-co-AA) hydrogel at both pH values (3.5 and 8.5) had lower equilibrium swelling degrees. The lyophilized p(NIPAM-co-AA) hydrogel absorbed solvent faster at 37 °C in the first 100 min of swelling due to larger pore size obtained during lyophilization.

The increase of temperature from 25 to 37 °C caused the contraction of non-lyophilized and lyophilized p(NIPAM-co-AA) hydrogel and the decrease of equilibrium swelling degree (Figure 9 and Figure 10), so this copolymeric hydrogel is classified as negative temperature-sensitive.

Values of kinetic parameters (*n*, *k* and *D*) for the swelling process of p(NIPAM-co-AA) hydrogel before and after lyophilization at 25 and 37 °C and pH values of 3.5 and 8.5 are given in Table 3.

The diffusion exponent *n* of the p(NIPAM-co-AA) hydrogel before lyophilization had a value of 0.822–1.014 (Table 3). The swelling of the p(NIPAM-co-AA) hydrogel before lyophilization was controlled by the solvent diffusion and the relaxation of polymer chains (anomalous diffusion mechanism). The exception was the swelling of the hydrogel at 25 °C and pH 3.5 which was controlled by the relaxation of polymer chains (Super Case II). After lyophilization, the swelling of the p(NIPAM-co-AA) hydrogel at all temperature values and pH was controlled by the solvent transport into the polymer matrix (“less Fickian” diffusion) (*n* < 0.5, Table 3).

The lyophilized p(NIPAM-co-AA) hydrogel had higher values of the diffusion coefficient *D* and constant *k* (Table 3), which indicated a higher degree of solvent penetration into the hydrogel. This result can be explained by the fact that lyophilized hydrogel has larger distances between the nodes of the polymer network, which enables faster solvent diffusion. 

In order to examine the influence of process factors (pH and temperature) on the system response (equilibrium swelling degree, α_e_) of p(NIPAM-co-AA) copolymer with 5 mol% of AA and 1.5 mol% of EGDM, nine experiments were performed.

The matrix of the full two-factor three-level experimental design (3^2^) with experimental values of the responses is shown in Table 4.

The quadratic model is better and more acceptable in comparison to linear and two-factor interaction (2FI) models for representing the influence of temperature and pH of the solution on the equilibrium swelling degree of the synthesized p(NIPAM-co-AA) hydrogel. The quadratic model has the highest value of coefficient of determination (*R*^2^ = 0.98) and adjusted coefficient of determination (adj. *R*^2^ = 0.95) and the lowest value of standard deviation. The results of the analysis of variance (ANOVA) for the quadratic model of the equilibrium swelling of p(NIPAM-co-AA) hydrogel are given in Table 5.

The quadratic model is statistically significant, because *p*-value is lower than 0.05 (Table 4, *p* = 0.0089). Both independent variables—temperature (*X*_1_) and pH (*X*_2_)—were statistically significant members of the model, as well as squared term of the pH variable ($X_{2}^{2}$), *p* = 0.0342. The final Equation with coded values (11) for quadratic model of equilibrium swelling of the p(NIPAM-co-AA) hydrogel is given as:(11)$$Y=194.2-45.47X_{1}+100.22X_{2}-36.03X_{1}X_{2}-32.00X_{1}^{2}-62.64X_{2}^{2}$$

When coded values were replaced by the actual ones, the second-degree polynomial Equation (12) was obtained:(12)$$\alpha_{e}=194.2-45.47t+100.22\mathrm{pH}-36.03t\mathrm{pH}-32.00t^{2}-62.64{\mathrm{pH}}^{2}$$

Equations with coded values were used to determine the variables’ effects on the system response. A higher absolute value of the regression coefficient indicates a greater influence of the corresponding variable on the system response. The sign in front of the regression coefficient determines the type of variable influence on the response. The positive sign indicates the positive effect of the variable on the system response, whereas the negative one indicates the negative effect [47,48]. It can be observed that values of pH (*X*_2_) have the highest regression coefficient (100.22, Equation (11)) and thus the greatest influence on the equilibrium swelling degree of the p(NIPAM-co-AA) hydrogel. According to the equation with coded values, temperature (*X*_1_) has less influence on the system response. Based on the sign in front of the regression coefficients, it can be concluded that increasing the pH value causes an increase in the equilibrium swelling degree of the p(NIPAM-co-AA) hydrogel, while temperature has the opposite effect. The functional dependence of the system response on variables is shown in Figure 11.

From Figure 11, it can be observed that with the increase of medium pH, the equilibrium swelling degree of p(NIPAM-co-AA) hydrogel increases. According to the model, the maximum value of the swelling degree of p(NIPAM-co-AA) hydrogel was obtained in the solution with pH 8.5 at 25 °C, α_e_ = 281.11, while under the same conditions, the experimental value was α_e_ = 269.32.

### 3.3. Examination of p(NIPAM-co-AA) Copolymer as a Matrix for Modified Release of Biochanin A

#### 3.3.1. Structural Analysis of p(NIPAM-co-AA) Copolymer with Incorporated Biochanin A

The copolymer with incorporated biochanin A was analyzed by the FTIR method together with biochanin A and the synthesized p(NIPAM-co-AA) copolymer (Figure 12).

In the FTIR spectrum of biochanin A, a strong, broad absorption band in the range of 3400–3200 cm^−1^ with maximum at 3261 cm^−1^ is assigned to valence vibrations of phenolic OH groups, ν(OH). The characteristic valence vibrations of the phenolic C–O bond, ν(C–O)Ar, gave astrong band in the range of 1260–1000 cm^−1^, which is located at 1176 cm^−1^ in the spectrum of biochanin A [14,49,50]. In-plane deformation vibrations of hydroxyl groups, δ(OH), occur in the range of 1500–1300 cm^−1^ and give a low-intensity band with maximum at 1323 cm^−1^ in the spectrum of biochanin A. The strong absorption band with maximum at 1661 cm^−1^ can be assigned to valence vibrations of carbonyl group, ν(C=O). The characteristic absorption bands at 1625, 1585 and 1515 cm^−1^ in the spectrum of biochanin A originate from valence vibrations of aromatic double bonds, ν(C=C)Ar, which is in accordance with the literature [49]. The asymmetric valence vibrations of ether C–O–C bond, ν_as_(C–O–C), give two strong bands in the range of 1275–1200 cm^−1^, and they are found at 1258 and 1237 cm^−1^ in the spectrum of biochanin A.

By incorporating biochanin A into the p(NIPAM-co-AA) copolymer, the formation of hydrogen bonds between phenolic OH groups of biochanin A (proton donor) with oxygen from C=O and C–O groups of the side chains of p(NIPAM-co-AA) copolymer (proton acceptor) is expected. Besides that, the C=O group of biochanin A can form hydrogen bonds with NH and OH groups of the side chains of the p(NIPAM-co-AA) hydrogel (proton donor).

In the FTIR spectrum of p(NIPAM-co-AA) copolymer with incorporated biochanin A, the band originating from valence vibrations of the OH group of AA is shifted by11 units to lower wavenumbers (3477 cm^−1^) compared to its position in the spectrum of p(NIPAM-co-AA) copolymer. The decrease of the valence vibrations’ frequency indicates participation of the OH groups in the formation of hydrogen bond, and the magnitude of this decrease is proportional to the strength of the formed bond. The maximum at 3323 cm^−1^ in the FTIR spectrum of copolymer with biochanin A originating from valence vibrations of N–H group, ν(N–H), is shifted by 25 units towards higher wavenumbers in relation to its position in the spectrum of p(NIPAM-co-AA) copolymer. The shifting of the maximum originating from deformation vibrations of N–H groups, δ(N–H), for 1 unit towards higher wavenumbers (1544 cm^−1^) in the spectrum of copolymer with biochanin A indicated that the N–H group participates in the formation of hydrogen bonds.

The position of the amide band I, ν(C=O), remained the same (1653 cm^−1^) after biochanin A incorporation into the p(NIPAM-co-AA) hydrogel, indicating that this group did not participate in the intermolecular interactions between biochanin A and p(NIPAM-co-AA) hydrogel [51].

The absorption bands of the valence vibrations of the C–O bond in the FTIR spectrum of copolymer with biochanin A at 1249 cm^−1^ (ν_as_(C–O)) and 1176 cm^−1^ (ν_s_(C–O)) are shifted by 7 and 4 units respectively, towards higher wavenumbers in relation to their positions in the spectrum of p(NIPAM-co-AA) copolymer. The maximum of the absorption band of carbonyl group, ν(C=O), at 1717 cm^−1^ is shifted by 3 units to lower wavenumbers in the spectrum of copolymer with incorporated biochanin A compared to its position in the spectrum of (NIPAM-co-AA) copolymer. The mentioned shifts also indicate the participation of C=O groups in the formation of intermolecular hydrogen bonds.

In-plane deformation vibrations, δ(OH), give one band with maximum at 1387 cm^−1^ in the spectrum of copolymer with incorporated biochanin A, which is shifted by 64 units to higher wavenumbers relative to its position in the spectrum of biochanin A, indicating that OH groups participated in the formation of strong intermolecular hydrogen bonds, which is in accordance with the literature data [49].

Based on the FTIR analysis, the structure of p(NIPAM-co-AA) copolymer with incorporated biochanin A and formed intermolecular hydrogen bonds between copolymer and biochanin A is given in Figure 13.

#### 3.3.2. Scanning Electron Microscopy Analysis

The morphology of the synthesized p(NIPAM-co-AA) hydrogel, biochanin A and the influence of the incorporated biochanin A on the morphology of hydrogel was examined by using the SEM method. The hydrogel samples were swollen to equilibrium and then lyophilized in order to understand the morphology better. The obtained SEM micrographs are shown in Figure 14.

The pore size of the synthesized p(NIPAM-co-AA) copolymer in the swollen state goes up to 100 μm. The pores are fairly uniform, and this structural organization of the polymer network corresponds to macroporous polymers and provides enough free space for incorporation of different molecules (Figure 14a). The incorporation of crystalline biochanin A (Figure 14c) into the p(NIPAM-co-AA) hydrogel affects the cross-sectional morphology of the hydrogel, making it less porous because the pores are filled with biochanin A molecules (Figure 14b). The results obtained by SEM analysis indicate incorporation of biochanin A into the p(NIPAM-co-AA) copolymer, which is in accordance with results obtained by FTIR analysis.

#### 3.3.3. The Loading Efficiency of Biochanin A into the p(NIPAM-co-AA) Hydrogel

The loading efficiency of biochanin A into the polymeric network of non-lyophilized and lyophilized p(NIPAM-co-AA) hydrogels was determined in relation to the total available mass of biochanin A (*L*_u_ = 60 mg/g_xerogel_). The masses of the non-lyophilized and lyophilized p(NIPAM-co-AA) hydrogels and incorporated biochanin A (*L*_g_), as well as loading efficiency (*η*), are shown in Table 6.

The presented results showed satisfactory loading efficiency of biochanin A into the polymeric network of non-lyophilized and lyophilized p(NIPAM-co-AA) hydrogels. The loading efficiency for lyophilized p(NIPAM-co-AA) hydrogel is higher (97.205%) compared to non-lyophilized hydrogels (92.767%), which is in accordance with the results of the swelling study.

#### 3.3.4. In Vitro Release of Biochanin A from p(NIPAM-co-AA) Copolymer

Release of biochanin A from non-lyophilized and lyophilized p(NIPAM-co-AA) copolymers was monitored in vitro at 37 °C and pH 4.5 and 7.9, which simulate body temperature and pH environment of the vaginal and rectumspace [51,52], using the UV/Vis method. Results of these studies are shown in Figure 15a,b, respectively.

Results of biochanin A release at 37 °C in a fluid at pH 4.5 for 12 h show that the released amount from non-lyophilized p(NIPAM-co-AA) copolymers is 24.82 mg/g_xerogel_ (41.37%), and from lyophilized 27.71 mg/g_xerogel_ (46.18%) of the total available amount (Figure 15a). In both copolymers, after 12 h, more than 50% of biochanin A remained in the pores, which provides the possibility of prolonged release in a medium that simulates the vaginal space.

The content of biochanin A released from a non-lyophilized copolymer p(NIPAM-co-AA) in pH fluid of 7.9 at 37 °C for 12 h is 50.57 mg/g_xerogel_ (84.28%), where the content of biochanin A released from the lyophilized copolymer is 53.29 mg/g_xerogel_ (88.83%) relative to the available amount, under pH conditions corresponding to the rectum.

Studies show that pH of the surrounding medium and lyophilization have an effect on the release of biochanin A from copolymer p(NIPAM-co-AA). From both copolymers p(NIPAM-co-AA), non-lyophilized and lyophilized, a higher amount of biochanin A (84.28–88.83%) is released at pH 7.9 than at pH 4.5 (41.37–46.18%), respectively. At a physiological body temperature (37 °C) that is higher than the volume phase transition temperature of lower critical solution temperature (LCST) [44,45] copolymer p(NIPAM-co-AA), the intermolecular interactions between biochanin A and side groups of the polymer matrix get broken and contraction of the polymer matrix starts, which initiates drug release.

Kinetic parameters of the release of biochanin A from the matrix of copolymer p(NIPAM-co-AA) calculated using Equations (6)–(9) (Table 7) show that the process flows according to “Less Fickian” diffusion law. The drug transport is slower than the polymer chain relaxation process and is controlled by the diffusion process.

Higher values of the diffusion coefficient, *D,* from the lyophilized hydrogel p(NIPAM-co-AA) indicate a higher rate of biochanin A release, which is expected due to the wider distance between the nodes of the polymer network.

According to the test results, it has been shown that copolymer p(NIPAM-co-AA), non-lyophilized and lyophilized, can be suitable as a carrier for modified release of biochanin A for rectal and vaginal application. Formulations of biochanin A with pH- and thermo-sensitive p(NIPAM-co-AA) copolymer, non-lyophilized and lyophilized, may be of interest for further testing.

## 4. Conclusions

The poly(*N*-isopropylacrylamide-co-acrylic acid), p(NIPAM-co-AA), copolymer was synthesized by free radical polymerization of *N*-isopropylacrylamide (NIPAM) monomer with 5 mol% of acrylic acid (AA) and 1.5 mol% of the cross-linker ethylene glycol dimethacrylate (EGDM) for alternative routes of biochanin A application. The increase of temperature caused the contraction of non-lyophilized and lyophilized p(NIPAM-co-AA) hydrogel and the decrease of equilibrium swelling degree, so this copolymeric hydrogel is classified as negatively thermo-sensitive. Based on the sign in front of the regression coefficients in the equation obtained by three-level factorial design, it can be concluded that increasing the pH value causes an increase in the equilibrium swelling degree of the p(NIPAM-co-AA) hydrogel, while temperature has the opposite effect. Since the total content of residual reactants is less than 1%, the synthesized copolymer p(NIPAM-co-AA) can be considered safe for use as a carrier of active substances. The FTIR analysis of the hydrogel with incorporated biochanin A indicated that the hydrogen bonds between polymeric chains and molecules of biochanin A are dominant. The pore sizes of the synthesized p(NIPAM-co-AA) copolymer in the swollen state were determined by SEM analysis of the lyophilized sample and went up to 100 μm, suggesting that synthesized hydrogel can be classified as macroporous. The loading efficiency of biochanin A into the synthesized hydrogel was up to 60 mg/g_xerogel_, and the release of biochanin A was faster at pH 7.9 than at pH 4.5. About 50% of the incorporated biochanin A was released from the lyophilized hydrogel at pH 7.9 and temperature of 37 °C in the initial 6 h. The obtained results indicate the possibility of using pH- and thermo-sensitive p(NIPAM-co-AA) copolymer as a carrier for modified release of biochanin A in the acidic environment of the vaginal cavity and weakly alkaline environment of the rectum.

## 5. Patents

Patent Application RS2021P0070, Gajić, I.; Ilić-Stojanović, S.; Dinić, A.; Zdravković, A.; Urošević, M.; Nikolić, L.; Nikolić, V.; Petrović, S.; Ilić, D. Formulations of biochanin A with pH- and thermo-sensitive copolymers, Priority 20 January 2021, the Intellectual Property Office of the Republic of Serbia.

## Figures and Tables

**Figure 1 polymers-13-00426-f001:**
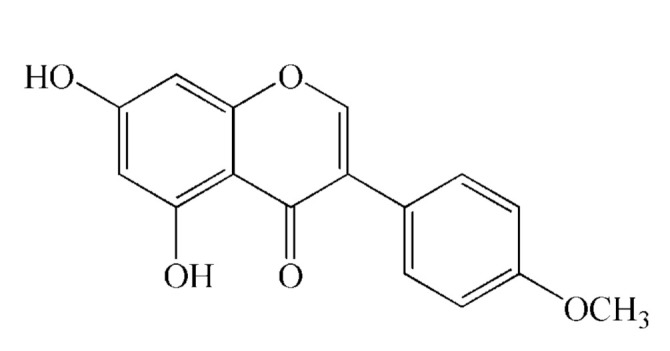
The chemical structure of biochanin A.

**Figure 2 polymers-13-00426-f002:**
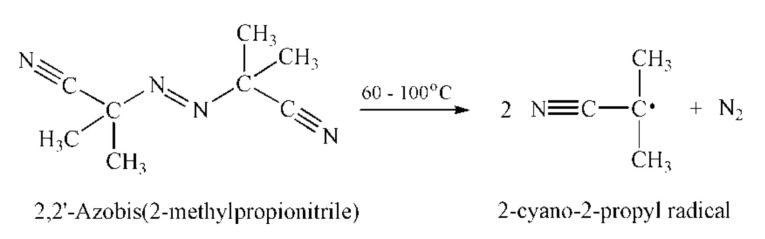
The formation of 2-cyano-2-propyl radical by degradation of 2,2’-azobis(2-methylpropionitrile).

**Figure 3 polymers-13-00426-f003:**
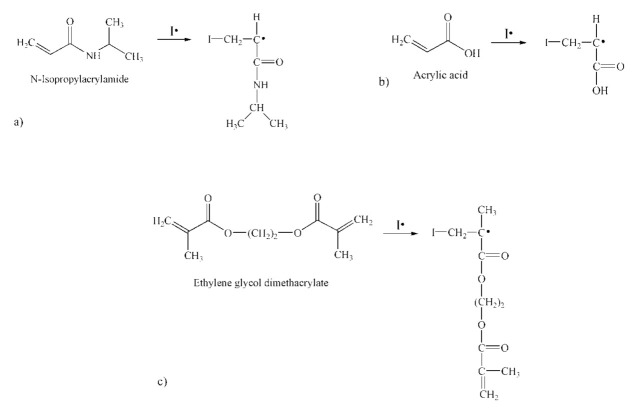
The formation of the radical species by reaction of the primary 2-cyano-2-propyl radical (I⋅) with: (**a**) *N*-isopropylacrylamide, (**b**) acrylic acid and (**c**) ethylene glycol dimethacrylate.

**Figure 4 polymers-13-00426-f004:**
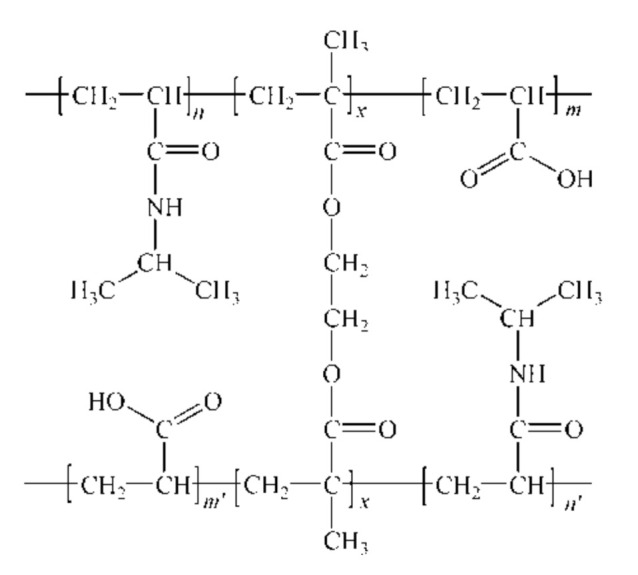
The possible structure of the synthesized copolymer poly(*N*-isopropylacrylamide-co-acrylic acid), p(NIPAM-co-AA).

**Figure 5 polymers-13-00426-f005:**
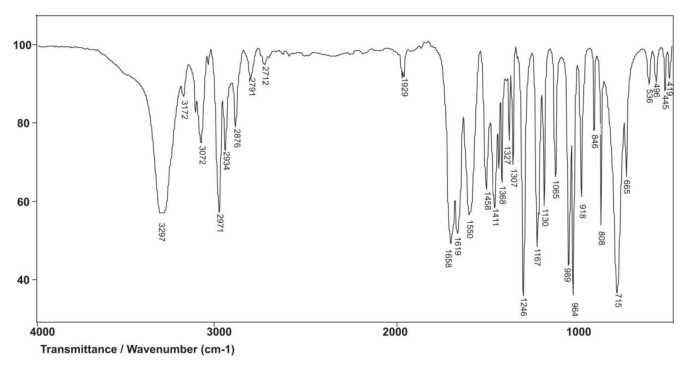
Fourier transform infrared spectroscopy (FTIR) spectrum of the monomer *N*-isopropylacrylamide, NIPAM.

**Figure 6 polymers-13-00426-f006:**
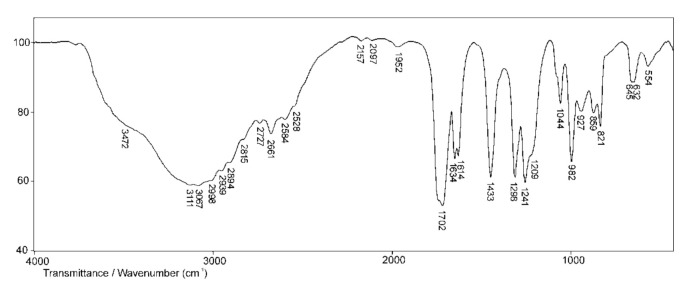
FTIR spectrum of the comonomer acrylic acid (AA).

**Figure 7 polymers-13-00426-f007:**
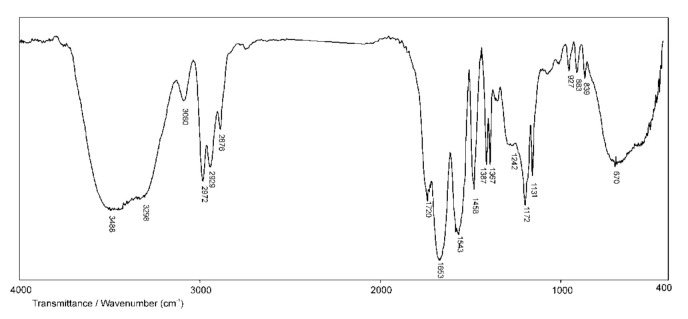
FTIR spectrum of the synthesized copolymer p(NIPAM-co-AA).

**Figure 8 polymers-13-00426-f008:**
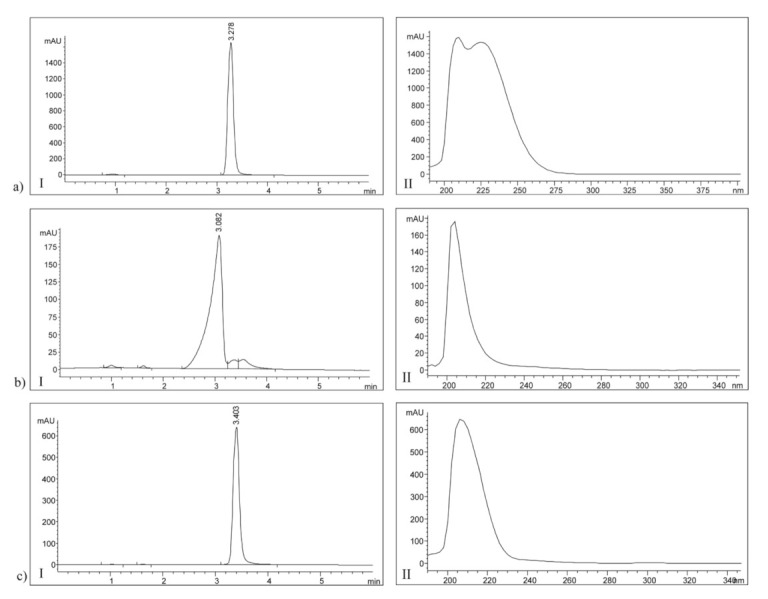
The high-pressure liquid chromatography (HPLC) chromatograms and UV spectra of standard solutions of (**a**) NIPAM, *c* = 0.253 mg/cm^3^, (**b**) AA, *c* = 0.150 mg/cm^3^ and (**c**) EGDM, *c* = 0.132 mg/cm^3^.

**Figure 9 polymers-13-00426-f009:**
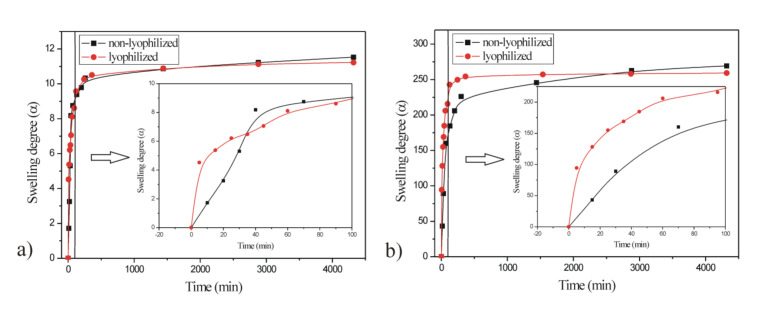
Dependence of the swelling degree of the p(NIPAM-co-AA) hydrogel before and after lyophilization on time at 25 °C and (**a**) pH 3.5 and (**b**) 8.5.

**Figure 10 polymers-13-00426-f010:**
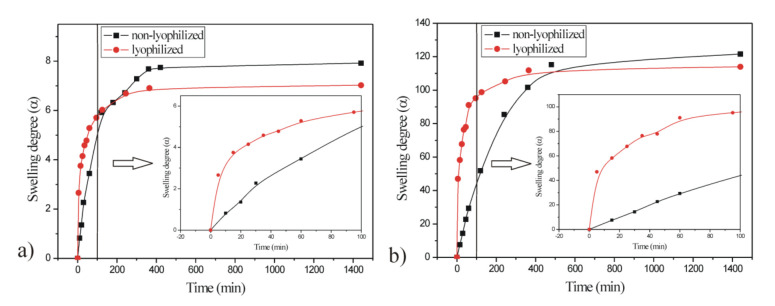
Dependence of the swelling degree of the p(NIPAM-co-AA) hydrogel before and after lyophilization on time at 37 °C and (**a**) pH 3.5 and (**b**) 8.5.

**Figure 11 polymers-13-00426-f011:**
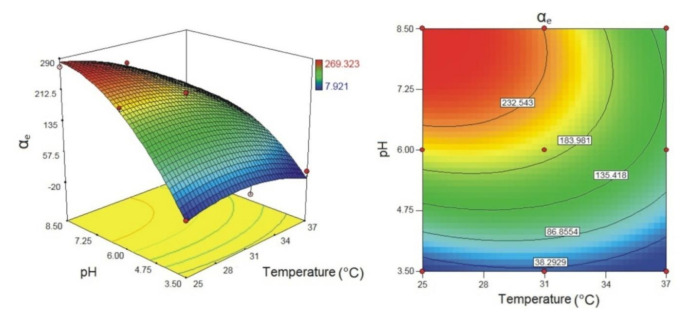
Response surface plot and contour plot for equilibrium swelling (α_e_) of p(NIPAM-co-AA) hydrogel as a function of pH and temperature.

**Figure 12 polymers-13-00426-f012:**
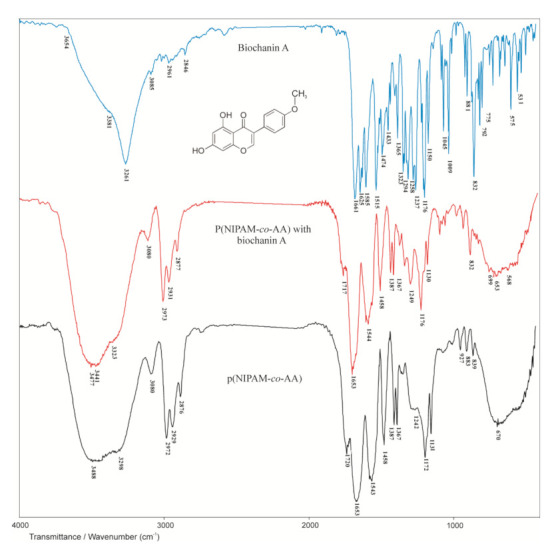
FTIR spectra of biochanin A, p(NIPAM-co-AA) copolymer with incorporated biochanin A and p(NIPAM-co-AA) copolymer.

**Figure 13 polymers-13-00426-f013:**
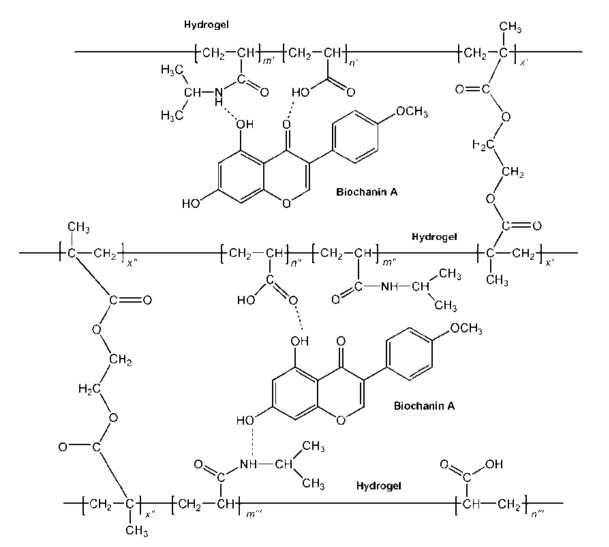
The structure of p(NIPAM-co-AA) copolymer with incorporated biochanin A.

**Figure 14 polymers-13-00426-f014:**
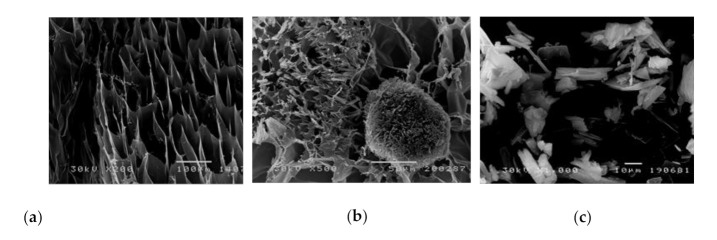
Scanning electron microscopy (SEM) micrographs of: (**a**) p(NIPAM-co-AA) hydrogel with 5% AA and 1.5 mol% of EGDM swollen to the equilibrium state with magnification 200× (scale bar 100 μm), (**b**) p(NIPAM-co-AA) hydrogel with 1.5 mol% of EGDM and incorporated biochanin A swollen to the equilibrium state with magnification 500× (scale bar 50 μm) and (**c**) biochanin A with magnification 1000× (scale bar 10 μm).

**Figure 15 polymers-13-00426-f015:**
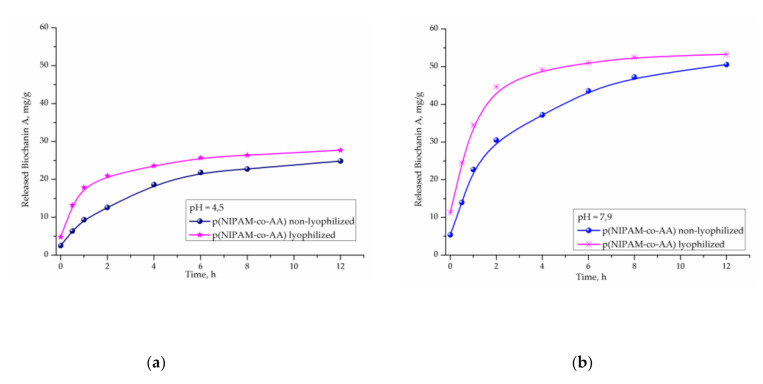
Released biochanin A content at 37 °C from non-lyophilized and lyophilized poly(*N*-isopropylacrylamide-co-acrylic acid), p(NIPAM-co-AA), copolymers, at: (**a**) pH = 4.5 and (**b**) pH = 7.9.

**Table 1 polymers-13-00426-t001:** Factors and levels in full three-level factorial design for swelling process of p(NIPAM-co-AA) hydrogels.

Factors	Coded	Actual	Level Values	
			Coded	Actual
Temperature (°C)	*X* _1_	*t*	−1 0 1	25 31 37
pH	*X* _2_	pH	−1 0 1	3.5 6.0 8.5

**Table 2 polymers-13-00426-t002:** The content of residual reactants in synthesized copolymer p(NIPAM-co-AA).

p(NIPAM-co-AA) Sample	NIPAM		AA		EGDM	
	mg/g	%	mg/g	%	mg/g	%
95/5/1.5	1.937	0.212	0.103	0.175	0.101	0.382

**Table 3 polymers-13-00426-t003:** Kinetic parameters of p(NIPAM-co-AA) hydrogel swelling before and after lyophilization at different pH values (3.5 and 8.5) and temperatures (25 and 37 °C).

Temperature (°C)	pH	Before Lyophilization				After Lyophilization			
		*n*	*k ×* 10^2^ (min^1/n^)	*R* ^2^	*D ×* 10^7^ (cm^2^/min)	*n*	*k* (min^1/n^)	*R* ^2^	*D ×* 10^4^ (cm^2^/min)
25	3.5	1.014	1.420	0.989	3.960	0.190	0.295	0.977	1.700
25	8.5	0.844	1.716	0.972	5.778	0.303	0.222	0.990	0.972
37	3.5	0.822	1.567	0.975	4.820	0.282	0.243	0.981	1.160
37	8.5	0.936	0.503	0.995	0.496	0.222	0.286	0.976	1.610

**Table 4 polymers-13-00426-t004:** Matrix of the full factorial design with experimental values of the responses.

Number of Experiment	*X*_1_ *t* (°C)	*X*_2_ pH	*Y*_exp._ α_e_
9	37	8.5	121.586
7	25	8.5	269.323
3	37	3.5	7.921
5	31	6.0	206.836
4	25	6.0	216.36
6	37	6.0	94.872
1	25	3.5	11.531
8	31	8.5	239.891
2	31	3.5	10.057

**Table 5 polymers-13-00426-t005:** Analysis of variance (ANOVA) test for quadratic model of equilibrium swelling of the p(NIPAM-co-AA) hydrogel.

Source	SS	df	MS	*F*	*p*	
Model	87,752.58	5	17,550.52	30.65	0.0089	significant
*X* _1_	12,406.49	1	12,406.49	21.67	0.0187	
*X* _2_	60,258.48	1	60,258.48	105.25	0.0020	
*X* _1_ *X* _2_	5193.15	1	5193.15	9.07	0.0571	
$X_{1}^{2}$	2047.47	1	2047.47	3.58	0.1550	
$X_{2}^{2}$	7847.00	1	7847.00	13.71	0.0342	
Residual	1717.66	3	572.55			
Cor Total	89,470.23	8				

SS—Sum of Squares, df—Degrees of Freedom, MS—Mean Square, *F*—Fisher distribution, *p*—Probability of error.

**Table 6 polymers-13-00426-t006:** The masses of xerogels and incorporated biochanin A (*L*_g_) and loading efficiency (*η*).

Hydrogel Sample	Mass of Xerogel (g)	*L*_g_ (mg/g_xerogel_)	*η*_biochanin A_ (%)
p(NIPAM-co-AA) non-lyophilized	0.0206	55.660	92.767
p(NIPAM-co-AA) lyophilized	0.0196	58.323	97.205

**Table 7 polymers-13-00426-t007:** Kinetic parameters of biochanin A release from poly(*N*-isopropylacrylamide-co-acrylic acid), p(NIPAM-co-AA), non-lyophilized and lyophilized, at pH values of 4.5 and 7.9 and at 37 °C.

Temperature (°C)	pH	Non-Lyophilized				Lyophilized			
		*n*	*k* (min^1/n^)	*R* ^2^	*D ×* 10^5^ (cm^2^/min)	*n*	*k* (min^1/n^)	*R* ^2^	*D ×* 10^4^ (cm^2^/min)
37	4.5	0.274	0.197	0.984	3.577	0.257	0.217	0.997	4.252
37	7.9	0.439	0.106	0.966	1.807	0.292	0.185	0.976	2.695

## Data Availability

No new data were created or analyzed in this study.

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
