# Peer review of "Modified Biochanin A Release from Dual pH- and Thermo-Responsive Copolymer Hydrogels"

_polymers, 2021, doi:10.3390/polym13030426_

Round 1

Reviewer 1 Report

I found the research in the paper well-conducted and interesting to read. The manuscript includes detailed experiments and conclusions about the use of p(NIPAM-co-AA) copolymer as a carrier for the further release of biochanin A content at 37°C.

I would like to suggest improving the scale bars in Figure 14 and the representation of the plots in figure 10. Also, I must require the use of a native English speaker to remove any grammatical errors in most of the paragraphs. It is obvious that the authors need some help polishing the paper. Just an example, the whole abstract is in the passive voice, which makes it very hard to read. Please, follow this advice and enhance the quality of the paper.

Reviewer 2 Report

This manuscript reported the work of incorporating biochanin A into the synthesized p(NIPAM-co-AA) copolymer and to examine its release at the body temperature and pH values that correspond to pH values of vaginal and rectal cavities. This manuscript verified the synthesis of the vector p(NIPAM-co-AA) through a variety of characterization methods, and systematically investigated different factors in the physicochemical and morphologically characteristics of the hydrogel. It has to be admitted that the workload is huge, I suggest that this article can be published after fully addressing several minor points as following:

  1. As mentioned in the author’s manuscript, several groups have prolonged the release of biochanin A and avoided extensive first-pass metabolism, so what is the significance of this work? Need to clarify this in the introduction part.
  2. The author speculated that the possible structure of the synthesized copolymer poly(N-isopropylacrylamide-co-acrylic acid) was shown in Figure 4 and Figure 7. We recommend author supplement the NMR data of the copolymers, which would be more beneficial to the structure speculation.
  3. Please also provide optical pictures of p(NIPAM-co-AA) hydrogel and p(NIPAM-co-AA) hydrogel incorporating biochanin A in the revised version.
  4. Please carefully check the format of all the reference, including the journal abbreviations and the volume and issue number.
  5. The authors could consider adding the following review articles and references which would again increase the interest to general functional hydrogel readers: Journal of Bioresources and Bioproducts, 2020, 5(3): 204-210; Biomaterials Science‚ 2020‚8, 4940-4950; Materials Horizons, 2020‚7, 746-761

Round 2

Reviewer 1 Report

The reviewers´ comments were addressed. Please, you can proceed with the paper in the current form.